# Specializing SAM: Online Adaptation of the Segment Anything Model for Interactive Segmentation in Uncommon Situations

## Abstract

Interactive segmentation is the task of segmenting an object with the help of user guidance. It is mostly used to create ground truth segmentation masks for object instances more efficiently. Recently, the Segment Anything Model (SAM) has been published to provide a foundation model for segmentation based on user-generated prompts. Despite being trained on the largest instance segmentation dataset to this date (SA-1B), we show that the model fails at the task of interactive segmentation when confronted with situations that do not comport with the initial training data. Such situations may however regularly occur when the model is used in practice. To alleviate these problems, we use the information that becomes available during the interaction to adapt the model to the dataset while being in use. In order to not impede any real time experience desirable to the user, we construct our method with the aim of minimizing computational overhead. In our experiments we will demonstrate the efficacy of the proposed adaptation method on twelve different datasets which are uncommon to SAM's initial training data, with four of them being medical segmentation datasets. With our method we are able to cause reductions of up to 16.93 percentage points in the $FR_{20}@85$ metric, and reductions of up to 18.43 percentage points in the $FR_{30}@90$ metric. Additionally, there is an improvement of up to 3.311 clicks in the $NoC_{30}@90$ on ten out of twelve datasets.

## 1 Introduction

Many computer vision systems need object segmentation masks for single images as training material. The development of such systems has especially been aided by the existence of large datasets for regular consumer images, such as COCO (Lin et al., 2014) and ADE20k (Zhou et al., 2017). Some segmentation tasks, however, need much more specific data. Example domains for such cases are sports (Ludwig et al., 2023a;b), agriculture (Roggiolani et al., 2023), medical image segmentation (Jha et al., 2020), and robotic vision (Zhuang et al., 2023).

The annotation of instance segmentation datasets usually incurs a high effort. Not only is there a large cost associated for human annotators, but in some difficult cases the creation of a high quality mask is a non-negligible problem. An example for this would be the annotation of mask polygons, when the object edges are finely jagged. In consequence, this lead to the development of interactive segmentation systems. Such systems receive a simple, low-effort user interaction to create masks. This usually happens in iteratively interactive contexts: The human refines computed masks by repeatedly interacting with the system, adding progressively more guiding interactions while inspecting the mask. This process goes on until the user is satisfied with the quality of the mask. In most cases, such interactions take the form of clicks, but scribbles, bounding boxes and coarse masks constitute usable forms of user guidance as well. The class agnostic nature of this task renders it viable for any kind of prompt. This property has been exploited to create a large foundation model which is capable of performing interactive segmentation, the Segment Anything model or SAM (Kirillov et al., 2023). While SAM is trained on the large SA-1B dataset, which has been published in conjunction with the model, a lot of practical scenarios require the creation of datasets for very specific tasks. This is for example the case in smaller companies that seek to

create datasets for the usage of in-house applications of computer vision, such as the automatization of processes. Here, only a small set of objects might be interesting to annotate.

In our paper we are going to view SAM in the light of the interactive segmentation task on scenarios which are considerably different from regular consumer images. This first and foremost means the usage of appropriate metrics: The first one is the Number of Clicks (NoC) we need to annotated an object mask, and the second one is the Failure Rate (FR) which tells us about the percentage of cases in which we fail to do so with a reasonable number of clicks. Out of these two, we regard the failure rate as the more crucial metric, since it informs us about the limits of the model's segmentation capabilities. Due to the difference of the test-time domains from the training domain, we are going to look at possibilities for test-time adaptation with the aim of reducing the models failure rate during usage.

Whilst the most straight-forward way of adapting a model would be fine-tuning, this strategy requires a pre-existing annotated dataset in the target domain before the model can even be used. We, however, constrain ourselves to techniques which only incur a negligible computational overhead, while using the information that becomes available during the interaction. In addition to the user-created clicks, which can be regarded as ground truth information for single pixels, we are going to use a pruned version of the resulting mask. The model can thus be used directly, while getting progressively better on the test-time domain. For the purpose of validating the techniques we are going to adapt SAM to miscellaneous rare situations, as well as medical image segmentation tasks. It should be noted that our method is not strictly dependent on SAM, and could be used for the adaptation of other foundation models. Our contributions can be summarized as follows:

1. We explore the performance of SAM as an interactive segmentation model on a variety of datasets which differ from regular consumer images.

2. We test the limit of SAM's segmentation capabilites, and show that the model displays a considerable failure rate on domains which are different from general consumer images.

3. We show possible adaptation schemes which lower the failure rate without incurring considerable costs. The low memory overhead and fast adaptation render the usage of our method effectively for free.

## 2 RELATED WORK

### 2.1 INTERACTIVE SEGMENTATION

Interactive Segmentation uses various kinds of user guidance, with clicks being the most popular annotation mode. Maninis et al. (2018) use four extreme points of the objects surface as guidance to segment the object. Li et al. (2018) proposes to generate various segmentation masks and use a network to choose the best among them. Zhang et al. (2020) combines bounding boxes with clicks on the object surface as user input. Dupont et al. (2021) use points at opposing sides of the object as segmentation guidance. The work of Sofiiuk et al. (2022) explores various input and training paradigms for interactive segmentation, whilst using the most recent mask as an additional form of input. The general training scheme is applied to networks with ViT-based backbones in Liu et al. (2022). Recently, Kirillov et al. (2023) have proposed the so called Segment Anything model (SAM) together with SA-1B, the largest interactive segmentation dataset to date containing over 1.1B segmentation masks. Due to the availability of the weights of the Segment Anything Model, there have been various papers which fine-tune its weights in order to adapt the model to a specific task. Cheng et al. (2023) and Wu et al. (2023) adapt SAM to various medical image segmentation tasks. Wang et al. (2023) use a modified version of SAM for robotic surgery. In Chen et al. (2023), adapter layers are introduced at intermediate places in the SAM-Encoder in order to fine-tune SAM to unusual image segmentation tasks. The method in Ding et al. (2023) adapts FastSAM (Zhao et al., 2023) for the task of change detection in remote sensing. It should be noted that all aforementioned methods require some additional fine-tuning on an existing annotated dataset in the target domain before they can be used. In contrast to that, our method can be used directly and adapts the network on-the-fly.

## 2.2 TEST-TIME ADAPTATION

The field of test-time adaptation deals with techniques to improve the model while it is already in use. Most of the existing methods are employed in contexts where there is no access to high-quality pseudo-labels, as would be the case in interactive segmentation. The method proposed by Wang et al. (2020) leverages entropy-minimization to adapt the model. Song et al. (2023) adds small blocks to the adapted models in order to enable test-time adaptation on low memory devices while using entropy-minimization as well. Wang et al. (2022) use a consistency loss and a exponential moving average, while stochastically restoring single weights to mitigate error accumulation. The methods most strongly related to this paper, are methods which focus on the adaptation of interactive segmentation models during usage. The most commonly exploited information in these methods are the user generated clicks. Albeit very sparse, they provide immediately available ground truth information. Kontogianni et al. (2020), Shi et al. (2023) and Lenczner et al. (2020) all exploit the clicks which are available due to the user interaction. The authors of Wang et al. (2018a) fine-tune their model on the basis of scribbles. The work of Hao et al. (2022) is most similar to our method, since the authors mention that they use intermediate masks, although they do not mention any method avoiding erroneous masks or regions. In contrast to our method, they also introduce additional modules to their model which requires an additional previous fine-tuning stage.

## 3 METHOD

### 3.1 PROBLEM STATEMENT

First, we will provide a precise description of the interactive segmentation problem. Afterwards, we will briefly describe how we simulate the interaction in order to test such a system. Assume that we have an image $x \in \mathbb{R}^{H \times W \times 3}$ and wish to create a segmentation map $m \in \{0, 1\}^{H,W}$ which delimits a desired area in said image. That is, every pixel belonging to the area in $x$ is set to $1$ in $m$, and every other pixel to $0$.

In order to create such an annotation, a user will repeatedly interact with a neural network $f_{\text{Seg}}$ by providing it with clicks that indicate pixels reliably belonging to the foreground or background of the image. In each step $t$ the user will be shown the current estimation of the mask $m_{t-1}$, which only consists of background pixels in the beginning ($t = 0$). The user then chooses a falsely labeled region from the mask and places a click $p_t$ on its surface. This $p_t$ is a triple $(i_t, j_t, l_t)$ which indicates a position $(i, j) \in \{1, ..., H\} \times \{1, ..., W\}$ and, depending on the choice of the user, a label $l \in \{+, -\}$ marking the position as foreground or background. The model $f_{\text{Seg}}$ is then given $m_{t-1}$, all previous clicked pixels $p_{1:t} = \{p_1, ..., p_t\}$ and the image $x$ in order to predict an improved mask $m_t = f_{\text{Seg}}(x, p_{1:t}, m_{t-1})$.

Once the user regards the quality of the mask as satisfactory, the interaction stops by saving this mask as $m^{\text{Res}}$, and the next image is annotated. It is to be noted that this result mask $m^{\text{Res}}$ might still be partially erroneous if the user chooses to ignore falsely annotated parts.

When it comes to evaluating the quality of such systems, we do not usually have a user at our disposal. Instead, we follow Sofiiuk et al. (2022) to simulate user interaction on images for which we already have ground truth segmentation masks $m^{\text{GT}}$. At each iteration, we first compute the false positive area $m_{\text{FP}}$ and the false negative area $m_{\text{FN}}$. Then we compute the euclidean distance transforms $\mathcal{D}(m_{\text{FP}})$ and $\mathcal{D}(m_{\text{FN}})$ of the respective error masks, and select the pixel with the largest value on both distance transforms as a click. The label of the click depends on whether it has been placed on $m_{\text{FP}}$ or $m_{\text{FN}}$. We stop the interaction once the overlap of the proposed mask $m_t$ with the ground truth mask $m^{\text{GT}}$ exceeds a certain minimum IoU. This final mask will then be treated as the result mask $m^{\text{Res}}$.

### 3.2 FOUNDATION MODELS FOR INTERACTIVE SEGMENTATION

The so called Segment Anything Model (SAM) is a large foundation model for the general task of *promptable segmentation*, which has been published in Kirillov et al. (2023) alongside the SA-1B dataset. Promptable segmentation denotes the task of segmenting arbitrary object instances as indicated by a user interaction, such as bounding boxes, text prompts or foreground/background clicks, as well as previously available low-quality masks. The ability to improve upon previous

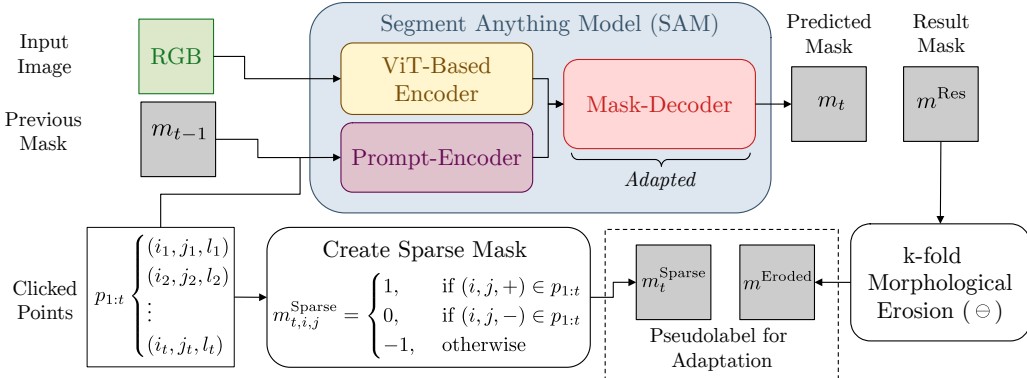

Figure 1: A rough description of the SAM architecture and the information used as pseudo-labels. Our method only adapts the mask-decoder which renders the computational effort of the backpropagation and optimization negligible. The usage of pseudo-labels is discussed in subsection 3.3.

masks and being guided by foreground/background clicks renders every promptable segmentation model compatible with click-based interactive segmentation. In addition to that, SAM has been pretrained on the SA-1B dataset, which contains 1.1B class-agnostic segmentation masks for 11M images. This causes SAM to be an extraordinarily good model for segmentation of objects on consumer images. Despite of this, there is still room for improvement when it comes to more specific image domains and more obscure types of objects, as our experiments indicate.

The architecture of SAM itself is divided into three parts: An *image encoder*, a *prompt encoder* and a *mask decoder*. The image encoder receives an image $x \in \mathbb{R}^{H \times W \times 3}$ and encodes it into a feature map independently of any user interaction. The authors of SAM use a ViT backbone for this task. The prompt encoder receives the prompt in the form of clicks, bounding boxes, and masks, and encodes them into a form which is useful for the mask decoder. The mask decoder receives the image features and the encoded prompts, and uses both to predict as segmentation mask for the object indicated by the prompts. Figure 3.2 contains a rough visualization the SAM architecture.

The greatest benefit of this general architecture lies in the decoupling of the computation of prompt embeddings and image features. The image only needs to be embedded once, while additional interactions only require a reuse of the prompt encoder and mask decoder. As long as the latter two networks are sufficiently light-weight, the user will be granted a real-time experience during the interactive usage of the model.

## 3.3 ADAPTING THE MODEL DURING TEST-TIME

When performing interactive segmentation, we generally annotate a sequence of images instead of just a single one. This opens up the possibility of exploiting information gathered from segmenting previous images, in order to get better at segmenting future images. Similar to Kontogianni et al. (2020) and Lenczner et al. (2020), we make use of the fact that each click on its own constitutes a single reliably correct ground truth pixel. Since this piece of ground truth is available directly after being entered by the user, we can already adapt the model while still annotating the image. Additionally, we use the mask $m^{\text{Res}}$ which results after the user is done annotating the image. We first subject the mask to multiple iterations of morphological erosion and then use this eroded mask $m^{\text{Eroded}}$ as a pseudo-label to adapt the model to the image domain.

When carrying out the adaptation, we only optimize the parameters of the decoder. A single execution of backpropagation and optimization with the Adam optimizer took 43.6 ms on a Nvidia V100 GPU vs. 13.1 ms for the corresponding forward pass, which is both much faster than a user could even consciously react to the mask. Since the accompanying optimization takes less than four times the time of the forward pass, the method doesn't impede any potential real time usage.

**Immediately using Clicks for Adaptation.** As soon as the user makes a click $\boldsymbol{p}_t = (i_t, j_t, l_t)$, we have ground truth information for a particular pixel at our disposal. We can use all clicks $\boldsymbol{p}_{1:t}$ we have received up until that point in order to create a sparse mask $\boldsymbol{m}_t^{\text{Sparse}}$ with

$$\boldsymbol{m}_{t,i,j}^{\text{Sparse}} = \begin{cases} 1, & \text{if } (i,j,+) \in \boldsymbol{p}_{1:t} \\ 0, & \text{if } (i,j,-) \in \boldsymbol{p}_{1:t} \\ -1, & \text{otherwise} \end{cases} \tag{1}$$

where $-1$ marks unknown pixels. Let $\boldsymbol{m}_t$ be the segmentation mask that has been computed after that last click has been made. We then compute a sparse binary cross entropy loss

$$\begin{aligned} \mathcal{L}_{\text{Sparse}}(\boldsymbol{m}_t^{\text{Sparse}}, \boldsymbol{m}_t) = &\frac{\sum_{i,j} \mathbb{1}_{\boldsymbol{m}_{t,i,j}^{\text{Sparse}}=1} \mathcal{L}_{\text{BCE}}(\boldsymbol{m}_{t,i,j}^{\text{Sparse}}, m_{t,i,j})}{\sum_{i,j} \mathbb{1}_{\boldsymbol{m}_{t,i,j}^{\text{Sparse}}=1}} \\ &+ \frac{\sum_{i,j} \mathbb{1}_{\boldsymbol{m}_{t,i,j}^{\text{Sparse}}=0} \mathcal{L}_{\text{BCE}}(\boldsymbol{m}_{t,i,j}^{\text{Sparse}}, m_{t,i,j})}{\sum_{x,y} \mathbb{1}_{\boldsymbol{m}_{t,i,j}^{\text{Sparse}}=0}} \end{aligned} \tag{2}$$

using $\boldsymbol{m}_t^{\text{Sparse}}$ as the label mask. We then immediately carry out an optimization step, thus progressively overfitting to the particular image as we continue annotating it. Note that this overfitting is deliberate and has to be reversed after we are done with the image. In order to achieve this, we reset the weights to their values before the image annotation, directly after we are done with the image.

**Using all Clicks to adapt the Model to the Image Sequence.** While the last paragraph describes a deliberate overfitting to the image, we also have the option to only carry out a single optimization step after we finish annotating the image. When doing this, we use all clicks that have been accumulated during the annotation of an image to create a single $\boldsymbol{m}^{\text{Sparse}}$ per image. The mask is created in the same fashion as before. This strategy adapts the model to the type of object and image domain of the test set, whilst acting less destructive on the parameters.

**Using the Resulting Mask to Adapt the Model to the Image Sequence.** Once the user regards the interactively created mask to be of sufficient quality, they stop the annotation and we obtain the result mask $\boldsymbol{m}^{\text{Res}} \in \{0,1\}^{H \times W}$. We can use this mask as a pseudo-label to adapt the model to the image sequence. In order to circumvent erroneous regions we will prune $\boldsymbol{m}^{\text{Res}}$ at the borders between foreground and background. This is done by separating the foreground and background masks, iteratively eroding both of them and uniting them again. Let $\boldsymbol{m}^{\text{FG}} = \boldsymbol{m}^{\text{Res}}$ and $\boldsymbol{m}^{\text{BG}} = 1 - \boldsymbol{m}^{\text{Res}}$ be the foreground and background masks, respectively. We define $\gamma^k(\boldsymbol{m})$ to be a $k$-fold application of morphological erosion as

$$\gamma^0(\boldsymbol{m}) = \boldsymbol{m}, \tag{3}$$

$$\gamma^{k+1}(\boldsymbol{m}) = \gamma^k(\boldsymbol{m}) \ominus \begin{bmatrix} 0 & 1 & 0 \\ 1 & 1 & 1 \\ 0 & 1 & 0 \end{bmatrix}, \tag{4}$$

where $\ominus$ is the symbol for the erosion operation. Then $\boldsymbol{m}^{\text{FG, Eroded}} = \gamma^k(\boldsymbol{m}^{\text{FG}})$ and $\boldsymbol{m}^{\text{BG, Eroded}} = \gamma^k(\boldsymbol{m}^{\text{BG}})$ are the eroded background and foreground masks. We will unite the two, resulting in the pruned pseudolabel mask $\boldsymbol{m}^{\text{Eroded}}$ with

$$m_{i,j}^{\text{Eroded}} = \begin{cases} 1, & \text{if } m_{i,j}^{\text{FG, Eroded}} = 1 \\ 0, & \text{if } m_{i,j}^{\text{BG, Eroded}} = 1 \\ -1, & \text{otherwise} \end{cases}. \tag{5}$$

We will carry out a single optimization step using $\mathcal{L}_{\text{Sparse}}$ after annotating each image.

**Using multiple decoders for Multiple Classes.** All of the aforementioned adaptation will inevitably overfit the model to a particular domain or type of object. It is however noteworthy, that the only part of the model to be adapted is the decoder. In cases where we want to annotate multiple different classes, we use multiple copies of the original decoder. Each of the copies is separately adapted to the respective object type or domain. We regard the memory overhead as negligible due

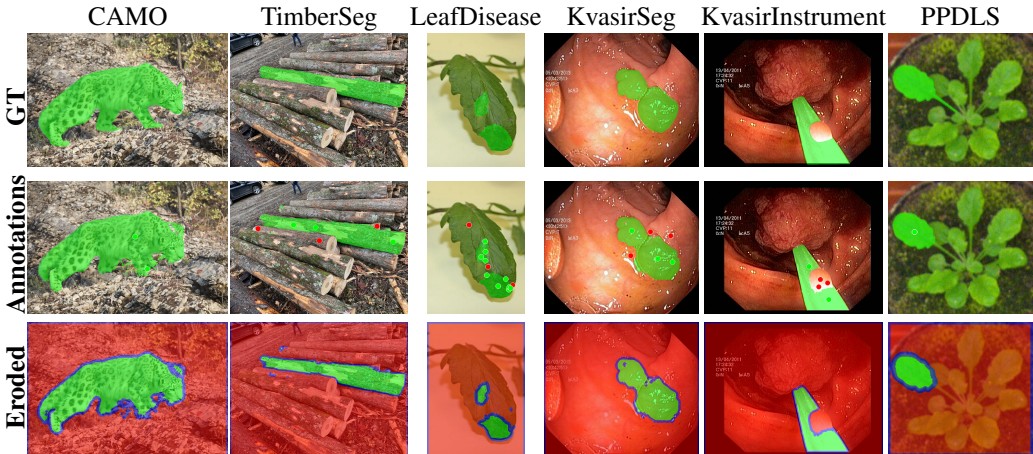

Figure 2: Examples for the masks occurring during the interaction. The *first* row contains the ground truth. The *second* row contains the annotated mask and the clicks. The *third* row contains examples for the eroded result mask. Green, red and blue correspond to foreground, background and the eroded area, respectively.

to the small amount of parameters of the decoder in comparison with the rest of the model. For the version of SAM with the ViT-b backbone, we have 4.06M parameters for the decoder vs 89.7M parameters for the rest of the model. For the versions with the ViT-l and ViT-h backbones, the rest of the model has 308.3M and 637M parameters respectively, while the decoder size remains the same.

# 4 EXPERIMENTS

## 4.1 EXPERIMENTAL SETTING

**Implementation Details.** During training we only adapt the decoder in order to minimize the computational overhead of our method. We carry out all optimization with a sparse binary cross entropy loss, as described in Section 3.3. We use the Adam optimizer with a learning rate of $10^{-6}$. The resolution of the input images is $1024 \times 1024$, which is a pre-existing property of SAM. All experiments use the ViT-b backbone. Whenever we use erosion, we carry out the iterative erosion with $k = 5$ iterations.

**Metrics.** When testing an interactive segmentation system, we want to exceed a certain IoU threshold $T_{IoU}$ within $n$ clicks. If the system is unable to do that, we consider the attempt at segmenting the image a failure and use $n$ as surrogate value for the number of clicks when computing the $NoC_n@T_{IoU}$. The *Number of Clicks* ($NoC_n@T_{IoU}$) metric measures the average number of clicks on the test set, while the *Failure Rate* ($FR_n@T_{IoU}$) measures the percentage of images on which the segmentation failed. Out of the two metrics we regard the failure rate as the more important one. While having to add an additional click on some images during the annotation process incurs a higher time effort, the failure rate measures the amount of images that cannot be segmented within a reasonable number of clicks at all.

**Adaptation Configurations.** When it comes to the techniques we employ during the adaptation, we can view them as configured in the following way:

**Click Adaptation (CA):** After each click, we can use all so far accumulated clicks to create a sparse label mask, with which we optimize the model to overfit to the image. We call this process *Click Adaptation (CA)*. In Subsection 3.3 we mentioned that this deliberate overfitting necessitates resetting the weight after each object, which we denote with an R for (R)eset in the tables. We may however choose to not perform this reset, and adapt our model continually over all images. We denote this by a C for (C)ontinual. No letter in the tables means that we do not use Click Adaptation at all.

Table 1: The results on datasets displaying rare objects types. NoC means the $NoC_{20}@85$ metric and FR is the $FR_{20}@85$, describing the number of objects that could not be segmented after 20 clicks. For both metrics, a smaller value indicates a better performance. An explanation of the configurations can be found in Section 4.1.

| Configuration | | | Rooftop | | DOORS | | TrashCan | | CAMO | |
|---|---|---|---|---|---|---|---|---|---|---|
| CA | RM | CM | NoC | FR | NoC | FR | NoC | FR | NoC | FR |
| | | | 4.171 | 6.00 | 5.439 | 16.69 | 13.259 | 57.42 | 7.224 | 20.3 |
| R | E | ✓ | 3.667 | 3.93 | 4.877 | 13.50 | 11.488 | 40.49 | 7.310 | 17.2 |
| R | | | 3.755 | 3.93 | 5.149 | 12.25 | 11.847 | 39.41 | 7.382 | 18.2 |
| C | | | 3.834 | 3.93 | 5.222 | 12.73 | 11.932 | 41.42 | 7.212 | 17.1 |
| | E | | 3.741 | 3.39 | 5.642 | 18.10 | 13.486 | 58.23 | 7.401 | 20.2 |
| | | ✓ | 3.915 | 4.62 | 5.154 | 14.97 | 13.694 | 59.47 | 7.278 | 19.4 |
| R | | ✓ | 3.707 | 3.70 | 5.326 | 12.83 | 11.796 | 40.38 | 7.402 | 17.0 |
| R | U | ✓ | 3.693 | 3.00 | 4.861 | 12.64 | 16.041 | 64.49 | 12.764 | 45.8 |
| Configuration | | | ISTD | | LeafDisease | | PPDLS | | TimberSeg | |
| CA | RM | CM | NoC | FR | NoC | FR | NoC | FR | NoC | FR |
| | | | 11.584 | 40.68 | 14.624 | 62.07 | 6.239 | 23.76 | 11.564 | 48.50 |
| R | E | ✓ | 10.392 | 31.13 | 14.595 | 60.71 | 6.250 | 20.04 | 10.497 | 39.67 |
| R | | | 10.932 | 34.66 | 14.665 | 61.05 | 6.267 | 19.25 | 11.080 | 42.26 |
| C | | | 10.896 | 33.91 | 14.631 | 60.71 | 6.218 | 19.43 | 10.661 | 40.73 |
| | E | | 11.295 | 38.80 | 14.690 | 61.05 | 5.955 | 21.42 | 10.745 | 43.32 |
| | | ✓ | 11.596 | 41.73 | 14.517 | 60.54 | 5.988 | 21.56 | 10.933 | 43.92 |
| R | | ✓ | 10.810 | 33.68 | 14.469 | 60.03 | 6.140 | 19.54 | 10.571 | 40.18 |
| R | U | ✓ | 15.017 | 57.97 | 14.918 | 62.41 | 14.387 | 49.40 | 16.710 | 74.76 |

**Result Mask (RM):** After being done with annotating an image, we can make use of the *Result Mask (RM)*. We could directly use the mask as a pseudolabel for optimization. We denote this with a U for (U)ntreated in the tables. As we will show however, this mask may still be erroneous and worsen our performance by subjecting our model to a partially false training signal. In order to circumvent this problem we may prune the masks foreground and background area by using iterative erosion. We denote this by an E for (E)rosion. No letter means that we do not make use of the result mask.

**Click Mask (CM):** After the annotation, we can use the accumulated clicks to form a sparse *Click Mask (CM)*, with which we can perform a single optimization step. In each configuration in which we do so, it is annotated by a checkmark (✓).

Specifically, these descriptions imply that the table row containing no letter or checkmark means that we are not performing any form of adaptation. This constitutes our baseline which is the regular SAM architecture. Whenever we use the Result Mask and the Click Mask in the same configuration, we just merge the two masks into a single mask. In all tables, the first line contains the baseline, while the second line contains our complete method. Figure 2 shows some qualitative examples.

### 4.2 ADAPTATION TO RARE OBJECTS

We will adapt SAM during usage on various datasets providing examples for rather obscure and uncommon situations. The Rooftop dataset (Sun et al., 2014) provides various remote sensing photos with annotated rooftops. The DOORS dataset (Pugliatti & Topputo, 2022) has been created for the segmentation of boulders. The TrashCan dataset (Hong et al., 2020) contains segmentation masks for underwater waste objects. CAMO (Le et al., 2019; Yan et al., 2021) is a dataset for the task of camouflaged object segmentation and ISTD (Wang et al., 2018b) for shadow segmentation. Additionally, we have three datasets for agricultural applications: One dataset for leaf disease segmentation (Alam, 2021), PPDLS (Minervini et al., 2016) for the segmentation of arabidopsis and tobacco leafs, and TimberSeg (Fortin et al., 2022) for the segmentation of logs in forestry work.

We are first going to look at $NoC_{20}@85$ and $FR_{20}@85$ metrics. According to Table 1, our method reduces the FR on ISTD from 40.68 to 31.13, while reducing the NoC by more than one click.

Table 2: The results on datasets displaying rare objects types. NoC means the $NoC_{30}@90$ metric and FR is the $FR_{30}@90$, describing the number of objects that could not be segmented after 30 clicks. For both metrics, a smaller value indicates a better performance. An explanation of the configurations can be found in Section 4.1.

| Configuration | | | Rooftop | | DOORS | | TrashCan | | CAMO | |
|---|---|---|---|---|---|---|---|---|---|---|
| CA | RM | CM | NoC | FR | NoC | FR | NoC | FR | NoC | FR |
| | | | 9.979 | 22.63 | 13.870 | 37.77 | 23.281 | 72.49 | 13.870 | 34.1 |
| R | E | ✓ | 8.891 | 18.21 | 13.163 | 33.62 | 20.527 | 54.06 | 13.488 | 28.3 |
| R | | | 8.961 | 18.24 | 14.996 | 36.30 | 20.979 | 53.86 | 13.719 | 29.6 |
| C | | | 9.358 | 19.86 | 14.623 | 35.35 | 21.032 | 53.40 | 13.573 | 29.1 |
| | E | | 9.321 | 19.63 | 14.965 | 42.47 | 23.700 | 73.30 | 14.082 | 33.0 |
| | | ✓ | 9.314 | 19.40 | 13.629 | 35.96 | 23.976 | 74.27 | 14.063 | 33.6 |
| R | | ✓ | 9.127 | 18.94 | 15.533 | 37.33 | 20.925 | 52.20 | 13.503 | 28.5 |
| R | U | ✓ | 9.339 | 19.40 | 13.082 | 33.31 | 25.221 | 70.75 | 20.840 | 54.2 |
| Configuration | | | ISTD | | LeafDisease | | PPDLS | | TimberSeg | |
| CA | RM | CM | NoC | FR | NoC | FR | NoC | FR | NoC | FR |
| | | | 18.744 | 49.02 | 24.255 | 72.62 | 13.260 | 38.55 | 20.358 | 62.64 |
| R | E | ✓ | 16.660 | 40.00 | 23.617 | 70.24 | 13.782 | 30.28 | 18.735 | 52.15 |
| R | | | 17.411 | 41.80 | 24.138 | 71.26 | 13.682 | 31.30 | 19.018 | 54.46 |
| C | | | 17.302 | 40.90 | 24.214 | 72.28 | 13.276 | 30.88 | 19.026 | 54.00 |
| | E | | 18.329 | 47.89 | 24.320 | 72.62 | 12.877 | 36.17 | 19.306 | 58.21 |
| | | ✓ | 19.574 | 53.08 | 24.226 | 71.60 | 12.574 | 35.07 | 19.436 | 58.76 |
| R | | ✓ | 17.217 | 41.35 | 24.153 | 72.11 | 13.447 | 31.22 | 18.874 | 53.49 |
| R | U | ✓ | 22.729 | 59.40 | 24.221 | 72.11 | 22.892 | 56.13 | 26.319 | 79.89 |

On TrashCan, our method even improves the FR from 57.42 to 40.49. It should also be noted that the results imply that SAM is unable to segment over half of the objects in the TrashCan and LeafDisease datasets to a satisfying degree. While our complete method slightly increases the NoC on the CAMO and PPDLS datasets, it still lowers the FR which we regard as the more crucial metric. In order to see the effect of using the untreated mask, we also run a version of our complete method without pruning the mask by erosion. As it turns out, eroding the mask is important due to potential erroneous areas at the edge of foreground and background area. The resulting false training signal manages to increase the FR by even more than two times on CAMO.

In Table 2, where the model needs to achieve an IoU of 90 within 30 clicks, we see an exacerbation of the problem SAM has with segmenting objects that are alien to its original training set. The FR values of the unadapted SAM model are 72.49, 72.62 and 62.64 on TrashCan, LeafDisease, and TimberSeg, respectively. This indicates that SAM is almost inept to segment these types of data to an IoU of 90 with the actual object surface, which would be considered necessary when producing annotations for new data. In the case of TrashCan and TimberSeg we manage to reduce the FR by 18.43 and 10.49 percentage points, respectively. The largest improvements regarding the NoC are incurred on TrashCan with a reduction of 2.754 clicks. On PPDLS, we again have a reduction in the FR for the cost of slightly higher NoC. It should be noted, that our complete method (CA = R, RM = E, CM = ✓) reduces the failure rate in all cases, and thus widens the applicability of SAM for uncommon domains.

### 4.3 RESULTS ON MEDICAL IMAGE SEGMENTATION

In order to investigate the efficacy of the adaptation method on medical image segmentation, we consider four different datasets: KvasirInstrument (Jha et al., 2021) contains segmented images of tools used in the gastrointestinal tract. CVCClinicDB (Bernal et al., 2015) and KvasirSeg (Jha et al., 2020) are two datasets for the task of polyp segmentation, while the GlaS dataset (Sirinukunwattana et al., 2017; 2015) provides data for the task of gland segmentation in colon histology.

The results for using our method on medical data generally comport with the results on other rare objects. It is first to be noted that our complete method causes a reduction of the failure rate in all cases. In Table 3 we see the complete method decreasing the FR on KvasirSeg from 2.7 to

Table 3: The results medical datasets. NoC means the $NoC_{20}@85$ metric and FR is the $FR_{20}@85$, describing the number of objects that could not be segmented after 20 clicks. For both metrics, a smaller value indicates a better performance. An explanation of the configurations can be found in Section 4.1.

| Configuration | | | KvasirInstrument | | CVCClinicDB | | GlaS | | KvasirSeg | |
|---|---|---|---|---|---|---|---|---|---|---|
| CA | RM | CM | NoC | FR | NoC | FR | NoC | FR | NoC | FR |
| | | | 2.137 | 1.86 | 4.935 | 8.17 | 7.485 | 14.64 | 3.615 | 2.7 |
| R | E | ✓ | 2.166 | 1.53 | 4.551 | 5.56 | 6.759 | 10.20 | 3.145 | 1.4 |
| R | | | 2.388 | 2.71 | 4.828 | 5.39 | 7.377 | 13.53 | 3.314 | 1.1 |
| C | | | 2.239 | 2.37 | 4.900 | 7.03 | 7.250 | 13.27 | 3.352 | 1.2 |
| | E | | 2.136 | 1.69 | 4.471 | 4.41 | 8.437 | 20.65 | 3.123 | 1.2 |
| | | ✓ | 2.178 | 2.37 | 4.637 | 5.39 | 8.539 | 20.72 | 3.281 | 1.2 |
| R | | ✓ | 2.305 | 2.37 | 4.757 | 6.21 | 7.576 | 15.29 | 3.273 | 1.0 |
| R | U | ✓ | 2.251 | 2.20 | 5.087 | 6.70 | 13.946 | 49.15 | 7.684 | 20.3 |

Table 4: The results medical datasets. NoC means the $NoC_{30}@90$ metric and FR is the $FR_{30}@90$, describing the number of objects that could not be segmented after 30 clicks. For both metrics, a smaller value indicates a better performance. An explanation of the configurations can be found in Section 4.1.

| Configuration | | | KvasirInstrument | | CVCClinicDB | | GlaS | | KvasirSeg | |
|---|---|---|---|---|---|---|---|---|---|---|
| CA | RM | CM | NoC | FR | NoC | FR | NoC | FR | NoC | FR |
| | | | 3.651 | 4.75 | 10.301 | 19.61 | 14.995 | 33.53 | 6.378 | 5.8 |
| R | E | ✓ | 3.825 | 4.58 | 8.585 | 10.46 | 11.684 | 19.15 | 5.580 | 3.9 |
| R | | | 4.063 | 5.42 | 9.343 | 14.05 | 13.341 | 24.12 | 6.397 | 5.7 |
| C | | | 4.041 | 5.42 | 9.041 | 12.75 | 13.331 | 23.73 | 6.057 | 4.4 |
| | E | | 3.749 | 5.08 | 9.588 | 14.87 | 15.884 | 35.49 | 5.573 | 3.4 |
| | | ✓ | 3.647 | 4.75 | 9.458 | 14.87 | 16.729 | 40.13 | 6.106 | 4.9 |
| R | | ✓ | 4.237 | 5.93 | 9.253 | 13.40 | 13.690 | 25.23 | 6.178 | 5.7 |
| R | U | ✓ | 4.239 | 5.76 | 12.446 | 21.57 | 22.744 | 55.29 | 16.168 | 34.2 |

1.4, almost halving it. On GlaS, the FR is lowered from 14.64 to 10.20 and the NoC is lowered from 7.485 to 6.759. On KvasirSeg and GlaS, the untreated result masks with a partially erroneous signal causes the most damage. It increases the failure rate by 18.9 and 38.95 percentage points in comparison to the full method with the eroded mask on each of the respective datasets. In Table 4, we can see a reduction in the FR by 14.38 percentage points, as well as a reduction in the NoC by 3.311 clicks on GlaS. On CVCClinicDB the FR is lowered by 9.15 percentage points, while the NoC is lowered by 1.716 clicks. On KvasirInstrument, the adaptation method causes a slightly higher NoC, but still lowers the failure rate.

## 5 CONCLUSION

In out paper we applied the Segment Anything Model to uncommon situations. We did so for the specific task of interactive segmentation and evaluated appropriate metrics: The Number of Clicks (NoC) and the Failure Rate (FR). Despite the model being trained on the largest dataset for instance masks to date, we see considerable problems when confronting the model with data that differs from regular consumer images. In some situations the model failed to segment more than half of the objects in the dataset, as reflected by the Failure Rate. This inability to segment certain objects poses a crucial limit to the model. In order to alleviate this problem we propose a test time adaptation method. All techniques are restricted to using information that occurs during usage and do not require any previous fine-tuning on existing datasets. In addition to that, they only incur a minimal computational overhead in order to not hamper any potentially required real-time capabilities. With the help of our method we manage to lower the Failure Rate on twelve different datasets and lower the NoC on ten of them. We thus conclude that the information available during test time provides a useful tool when applying a foundation model such as SAM to uncommon domains.

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
