# OpenReview forum: "Specializing SAM: Online Adaptation of the Segment Anything Model for Interactive Segmentation in Uncommon Situations"
_ICLR.cc/2024/Conference — ICLR 2024 Conference Withdrawn Submission_

### Official Review · Reviewer_onHB · 2023-10-25

**Soundness:** 2 fair
**Presentation:** 2 fair
**Contribution:** 1 poor
**Rating:** 1
**Confidence:** 5

**Summary:**

The paper discusses the limitations of the Segment Anything Model (SAM) in interactive segmentation when faced with situations not encountered during its initial training on a large dataset. To address these issues, the authors propose an adaptation method that uses interaction data to fine-tune the model in real-time, demonstrating its effectiveness on twelve diverse datasets, including medical segmentation data, where it achieves significant reductions in segmentation errors and improved user experience with reduced user input clicks.

**Strengths:**

This paper has achieved a test adaptation with relatively few changes (modify only the decoder section)to the existing SAM model.

**Weaknesses:**

1. There is no description of SOTA, and no performance comparison to SOTA.
2. The techniques (CA, RM, CM & table 1, 2) description lacks evidence or ablation experiments.
3. The performance improvement is not significant compared to the additional computation at the mask decoder stage. (table 1, 2) (I don't see why this model should be used?)

**Questions:**

Refer to Weaknesses

---

### Official Review · Reviewer_sPPi · 2023-10-27

**Soundness:** 2 fair
**Presentation:** 3 good
**Contribution:** 2 fair
**Rating:** 1
**Confidence:** 4

**Summary:**

This paper introduces a user-click-based adaptation method for test-time fine-tuning of SAM. This method leverages user clicks in several ways, encompassing single-click-based overfitting, model adaptation with a sparse mask defined by single-image clicks, and model adaptation with a sparse mask derived from each click. The authors have conducted comprehensive experiments on 12 datasets that are uncommon to SAM's initial training data. The results demonstrate the effectiveness of these fine-tuning strategies in significantly reducing the failure rate, even with a limited number of user clicks.

**Strengths:**

The paper assesses the impact of their model adaptation strategy using two key metrics: the Number of Clicks (Noc) and Failure Rate (FR). These metrics offer a user-centric perspective on interactive performance by gauging how effectively the failure rate can be reduced in SAM with a limited number of user clicks. The proposed method in this paper was tested on a diverse range of datasets, including remote sensing images, wastewater objects, and medical images, among others, consistently demonstrating improvements.

**Weaknesses:**

a. Novelty of this work. The authors assert that their method allows adaptation to the target domain dataset without requiring fine-tuning on an existing dataset. However, the use of online-clicks-based masks, including RM and CM, essentially resembles a form of fine-tuning on the target domain dataset. Notably, all experiments are conducted in an isolated fashion, without any comparisons to other adaptation methods. From a technical perspective, RM or CM could be considered as pseudo labels within the target domain dataset, which might be adaptable to other existing methods. As cited in section 2.2, several methods make use of immediate masks for SAM model adaptation. A more comprehensive comparison with state-of-the-art methods is needed to illustrate the advancement of this paper.

b. The motivation to introduce RM and CM for model adaption is not clear. As claimed in the paper, the interactive erosion step is performed to prune the foreground and background areas on the resulting mask and reduce false training signal. However, we notice that the experiment results contradict with this point. For examples, in Table 1, the eroded RM increase the NoC and FR on datasets DOORS and TrashCan. In Table 4, the eroded RM increase the NoC and FR on datasets KvasirInstrument Glas. The CM impede the performance for TrashCan in Table 1, ISTD in Table 1 and Table 2, KvasirInstrument in Table 3, and Glas in Table 4. From these results we could draw the conclusion that the introduction of RM and CM for model adaption is not that reasonable. From the other aspect, with comparison of row 2 and last row of all tables, we could observe a great reduction in failure rate. However, when taking row 3 and row 5 together into consideration, we found that the pruning step is actually cutting the error caused by RM in some extent. For example, in Table 1, with only CA (row 3), the NoC and FR are even smaller than using eroded RM (row 5).

c. Further clarification is warranted on certain points. Firstly, it is advisable to include an experiment setting that exclusively utilizes the uneroded RM recommendation. This addition would shed light on the impact of the interactive erosion process. Moreover, it is of utmost importance to conduct comparisons with other model adaptation methods applied to SAM. This comparative analysis is crucial for providing additional insight into the effectiveness of the method proposed in this paper.

**Questions:**

The proposed adaptation strategy essentially defines "The strength is that the simple and on-line adaption strategy shows considerable improvements on extensive different domain object segmentation." However, the experiment results in somewhere show contradiction with the proposed bags of adaption strategy, especially the RM and CM. The motivation to use eroded resulting mask as pseudo label for model fine-tuning is not clear as the experiment results indicate it is not the key factor improving the adaption effect, and even raises the failure rate sometimes.

---

### Official Review · Reviewer_eT7h · 2023-10-31

**Soundness:** 3 good
**Presentation:** 2 fair
**Contribution:** 2 fair
**Rating:** 3
**Confidence:** 4

**Summary:**

The paper proposes to finetune the mask decoder of the Segment Anything Model during the deployment as an interactive segmentation model. While SAM was trained on many images, not all domains of images are covered well in the Segment Anything dataset and as such it is to be expected that the interactive segmentation quality is not equal in all cases. The proposed fine-tuning method uses intermediate clicks to finetune the SAM mask decoder module, as well as a final fine-tuning with the eroded output mask. In most cases the method shows significant improvements, allowing more images in several datasets to be successfully segmented, while often also reducing the number of clicks needed.

**Strengths:**

- SAM is currently a very popular model and the paper proposes a fairly simple approach to improving it for interactive instance segmentation, while remaining real-time capable.
- Quite significant empirical improvements are shown.
- Most parts of the paper are reasonably well written and easy to follow.

**Weaknesses:**

- My main concern is the usage of SAM and the fact that the paper implies it's the new state-of-the-art interactive segmentation model. Sadly, the SAM paper is written in a somewhat misleading way when it comes to interactive segmentation. As the authors also noticed, in many cases SAM is not capable of arriving at high IoU segmentation masks. While the SAM paper also reads a bit as if it is a new state-of-the-art model with respect to interactive segmentation, it does not follow the standard interactive segmentation evaluations that are performed in most other works (e.g. NoC, as also used in this paper). In my own experience, SAM actually does not outperform recent methods when evaluated with standard metrics. I thus find this paper pretty misleading, given that it reads as if SAM is clearly the best existing interactive segmentation model, especially with the whole arguments about how much data SAM was trained. To me it is not clear at all why SAM thus is an obvious relevant baseline and how the resulting performance compares to other state-of-the-art methods. It should also be noted that while SAM was indeed partially trained for the interactive segmentation task, it was not the main focus, clearly highlighting that just using SAM for this task will most likely give less than optimal results.

- It is a little hard to find out what the exact novelty is supposed to be. As far as I can tell the click adaptation is not new, as well as the adaptation of a model across a sequence of images (both already discussed in Kontogianni et al. 2020). The usage of multiple decoders for multiple classes might be new, but that is not discussed anymore at all in the further paper. That would only leave the usage of eroded masks as a novelty? Which, while effective, is also not super surprising.

- The paper also states that the method does not actually depend on using SAM, but that the proposed method could also be applied to other models as well. So if the paper is really about the proposed method, it should for sure be evaluated with other base approaches as well, but this will not be trivial given that most methods don't have a small mask decoder that could be finetuned on the fly. On the other hand, if the main claim of the paper would actually be the high scores, it would actually need a few other interactive segmentation approaches evaluated as baselines on the used datasets.

- The related work and general citations are lacking. Looking at the interactive segmentation section, the paper gives the impression that there is a little handful of existing interactive segmentation methods and that SAM simply is the new state-of-the-art in this field. While RITM and SimpleClick are cited, the recent strong FocalClick method is missing. Looking at the related work sections of any of these three papers instantly shows there is a wealth of other interactive segmentation methods. Also, while we of course all know the Adam optimizer and ViTs, it would still be nice to give credit and actually cite the papers. And in general I find the citation density to be fairly low. For example the problem statement, which is definitely not new, has no citations.

**Questions:**

- The result tables now all list the same 8 settings. But some interesting ones are actually missing. First of all, what happens if you don't use click adaptation, but you do use the eroded mask as well as click masks. This is a clear version that is missing compared to the final proposed setting. Furthermore, there is only a single setting with continual click adaptation. In most cases this is actually better, when compared to the version where you reset the method after each image and don't do anything else. This makes me wonder if continual adaptation wouldn't actually perform quite a bit better than the final approach, when paired with eroded result masks, as well as click masks.

- SAM has two important features that are not discussed in this paper. First of all it predicts several masks for each query, one generic mask, as well as three masks of different hierarchical levels with predicted IoUs. Which of those do you use? According to the SAM paper they use the mask with the highest IoU prediction during training. When not using the same setup, I would not be surprised if you get subpar performances. Additionally, when reading the training section in the original SAM, it also discusses additional iterations without new clicks, which could be used to improve the resulting mask. As far as I can tell these are ignored in this paper?

- The paper states that you use a separate mask decoder per class, but typically interactive segmentation deals not only with semantic, but also with instance segmentation. If we now want to for example segment multiple people in an image, one can't use the proposed method to tune the model in a sequence, given that every person will be both positive and negative depending on which person is being annotated. Does this mean one would have to reset the encoder after each instance? And in that case, what are the separate mask decoders used for?

- Is there some source for the claim that 13.1ms and 43.6ms are both "much faster than a user could even consciously react to the mask"?

- Is there a difference between applying the used erosion operator k times vs simply applying a larger erosion kernel?

- For the continual adaptation, I would consider that the dataset order might actually have an effect on the performance. Did you run this multiple times to get an estimate of how stable this actually is?

- It might have been good to provide some statistics about the used datasets to easily estimate how relevant these are. Also the choice of datasets is not completely obvious. Some of those were also already used in the original SAM evaluation, while others are new?


Typos and small bugs:
"Figure 3.2" doesn't exist, also there is a missing "of" in that sentence.

---

### Official Review · Reviewer_WW8e · 2023-11-01

**Soundness:** 3 good
**Presentation:** 2 fair
**Contribution:** 3 good
**Rating:** 5
**Confidence:** 3

**Summary:**

This paper points out limitations of SAM when exposed to rare cases. The author propose to adapt SAM by using the information that becomes available during the interaction, to achieve progressively better performance on the test-time domain, with a manageable computational overhead. A series of experiments have illustrated that the proposed adaption method lowers the failure rate without incurring considerable costs.

**Strengths:**

The overall structure of the paper presentation is clear and easy to follow.

This paper targets at one limitation of SAM when it fails at segmenting rare cases. The proposed adaption solution is straightforward and intuitive.

The design for reducing computational overhead is useful.

The authors have conducted extensive experiments to illustrate the adaption's effectiveness.

**Weaknesses:**

While comparing to its baseline SAM, the authors have not provide discussions and comparisons against other interactive segmentation methods. It is not clear how effective the adaption method comparing to existing approaches.

There are many typos / grammar errors throughout the paper, just name a few: 'It is first to be noted that... ', 'masks ... causes ...' in Sec. 4.3; 'in out paper' in Sec. 5.

[1] SimpleClick: Interactive Image Segmentation with Simple Vision Transformers, ICCV, 2023
[2] A hybrid propagation network for interactive volumetric image segmentation, MICCAI, 2022
[3] Interactive object segmentation with inside-outside guidance, CVPR, 2020

**Questions:**

See weaknesses.